# Metabolomics Revealed the Effects of *Momordica charantia* L. Saponins on Diabetic Hyperglycemia and Wound Healing in Mice

**DOI:** 10.3390/foods13193163

**Published:** 2024-10-04

**Authors:** Dengdeng Zhao, Zixuan Luo, Shasha Li, Shuwen Liu, Chan Wang

**Affiliations:** 1Guangdong Provincial Key Laboratory of New Drug Screening, School of Pharmaceutical Sciences, Southern Medical University, Guangzhou 510515, China; 2State Key Laboratory of Organ Failure Research, Guangdong Provincial Institute of Nephrology, Southern Medical University, Guangzhou 510515, China

**Keywords:** *Momordica charantia saponins*, diabetes mellitus type 2, wound healing, metabolomics

## Abstract

*Momordica charantia* L. saponins (MCS) may promote wound-healing properties but the underlying mechanisms are unclear. This study aimed to examine the effects and mechanisms of MCS on diabetic wounds. The results have shown that higher MCS intake lowered fasting blood glucose levels, serum lipids, and lipopolysaccharides in diabetic mice. MCS-treated diabetic mice exhibited faster wound healing than the diabetic control groups. After three days, the diabetic control groups exhibited a wound area reduction of only 19.3%, while a 39.75% reduction was observed following high-dose MCS treatment. Five potential biomarkers were screened in the metabolomics study. The results revealed that MCS mainly regulated glycerophospholipid metabolism, fructose and mannose metabolism, steroid hormone biosynthesis, pyrimidine metabolism, and the Krebs cycle, thus affecting wound healing. Overall, MCS could not only exert a hypoglycemic effect but also promote diabetic wound healing, making it a potential treatment option for diabetes-related wounds.

## 1. Introduction

Diabetes mellitus is a prevalent and severe metabolic condition affecting over 463 million people globally, with an anticipated increase of 10.02% by 2030 [1]. Foot ulcers are the primary complication in patients with diabetic patients due to chronic hyperglycemia, impeding the healing process. Approximately 19–34% of diabetic patients develop diabetic foot ulcers during their lifetime, which is the primary cause of hospitalization [2]. Diabetic foot problems are not only vastly prevalent and expensive but are also associated with higher morbidity and mortality. Considering the substantial impact of diabetic foot ulcers on patient outcomes and healthcare costs, it is critical to explore novel approaches and strategies to enhance diabetic wound healing. Although current treatments for diabetes and its complications primarily depend on synthesized medications, traditional medicine in various countries has also acknowledged the beneficial effects of local plant resources in managing diabetes.

In contrast to synthesized pharmaceuticals, traditional Chinese medicine offers gentler and more enduring effects on regulating physiological functions. As a result, traditional Chinese medicine is viewed as a natural and safe therapeutic option [3,4]. Given these benefits, exploring traditional Chinese medicine as a treatment for patients is a promising and worthwhile approach. As a common traditional Chinese medicine, *Momordica charantia* L. (MC) has been extensively used in China for its antidiabetic, anti-inflammatory, antibacterial, antioxidant, and anti-ulcer activities [5]. MC primarily consists of proteins, polysaccharides, flavonoids, saponins, fatty acids, and volatile constituents [6]. *Momordica charantia* L. saponins (MCSs) are the main bioactive constituents of MC that have been demonstrated to improve glucose homeostasis and reduce insulin resistance [7]. Dend et al. affirmed that MCSs significantly diminished the degenerative alterations in islet β cells and bolstered reparative effects on the histological structure and the insulin secretion of the pancreas [8]. In another study, MCSs regulated the insulin signaling pathway by upregulating the expression of p-IRS-1 and p-Akt while downregulating p-IRS-1 [9]. In recent studies, researchers have examined the potential benefits of MC extracts in facilitating the process of wound healing [10,11,12]. When used topically on normal and diabetic wounds, MC extracts accelerated wound healing by increasing the skin protein content [10]. In a wound chamber model of diabetic rats, the local application of MC extracts accelerated and improved wound healing by preventing the regression of granulation tissue and blood vessels [11]. Further studies have confirmed that MC extracts have increased the phosphorylation of extracellular signal-regulated kinase (ERK)1/2 to promote angiogenesis [12]. Yet, as a traditional medicine and vegetable, the effects of oral administration of MC extracts on diabetic wounds have not been described.

Metabolite analysis provides a thorough understanding of the final products in the body and has been extensively employed in predicting occurrences, discovering biomarkers, and offering insights into underlying mechanisms [13]. Type 2 diabetes mellitus (T2DM) is a chronic metabolic disease associated with the functional decline of organs. In this context, metabolomics should be a favored tool for studying T2DM and its complications. In this study, we investigated and evaluated the effect of MCS on diabetic blood glucose and wound healing through oral administration. Moreover, we studied the mechanism of MCS in hypoglycemic activity and promoting wound healing in diabetic rats using non-targeted metabolism technology. These findings are expected to validate the wound-healing ability of MCS in diabetic patients and offer novel strategies for preventing and treating diabetic foot ulcers.

## 2. Materials and Methods

### 2.1. Materials and Reagents

MCS was purchased from Xi’an Wenzhou Biotechnology Co. Ltd. (Xi’an, Shanxi Province, China), containing saponins and protein contents of 68.63% and 5.37%, respectively. Streptozotocin (STZ) was obtained from Sigma-Aldrich Corporation (St. Louis, MO, USA) to create a diabetes model. Metformin was acquired from Bristol-Myers Squibb (Shanghai, China) and recognized as a positive control. Lipid detection kits were purchased from Nanjing Jiancheng Bioengineering Institute (Nanjing, Jiangsu Province, China) for testing serum lipids. Insulin and Lipopolysaccharides (LPS) ELISA kits were supplied by the Shanghai Enzyme-linked Biotechnology (Shanghai, China). L-2-chlorophenylalanine, acetonitrile, formic acid, isopropanol, and water for liquid chromatography/tandem mass spectrometry (LC–MS/MS) were obtained from Thermo Fisher Scientific Inc. (Waltham, MA, USA). The other chemical reagents that were used in the study were of analytical grade.

### 2.2. Animal Experiments

Animal. Male C57BL/6J mice (aged 4–5 weeks) were purchased from the Guangdong Medical Animal Experimental Center (Guangzhou, Guangdong Province, China). All the animals were housed in specific pathogen-free conditions without restrictions on food and water intake. The Animal Research Ethics Committee of Southern Medical University approved the protocol used in this study (**SMUL202311029**). Adequate measures were taken to minimize the pain of experimental animals.

STZ-induced diabetic mice. After one week of acclimatization, the mice were fed a high-fat diet for six weeks and then injected with STZ (60 mg/kg) intraperitoneally for four days. After seven days, the fasting blood glucose (FBG) levels were measured by Accu-Chek Performa (Roche Diagnostics, Mannheim, Germany). The mice were considered T2DM models if the FBG exceeded 11.1 mM [14]. A total of six groups (n = 8 per group) were used for the experiment: NC: normal control (saline); DC: diabetic control (saline); LMCS: low-dose MCS (50 mg/kg MCS); MMCS: medium-dose MCS (100 mg/kg MCS); HMCS: high-dose MCS (200 mg/kg MSC); MET: metformin (250 mg/kg). All groups were gavaged once daily for four weeks. The serum samples were collected in the fourth week of administration.

Establishment of skin wound model. Excisional dorsal skin wounds were made using a 6 mm hole punch [15]. After establishing the diabetic mice wound model, wound images were collected on days 0, 3, 7, and 10. The mice were sacrificed on day 10, and their wound tissues were collected for the experiment. The following formula was used to assess the wound healing rates in mice: Wound closure (%) $=\frac{{\;(A}_{0}–A_{t})\;}{A_{0}}\times 100$ (A_0_: Area of trauma on day 0; A_t_: Area of trauma on days 3, 7, and 10).

### 2.3. Analysis of FBG, Oral Glucose Tolerance Test, Insulin, and LPS

The FBG was examined using a glucose meter. An oral glucose tolerance test (OGTT) was conducted, and the serum glucose levels in the blood samples from tail veins were estimated using a glucometer at 0, 30, 60, and 120 min, and the areas under the glucose-time curve (AUC) were calculated as described previously [8]. The levels of insulin and LPS were detected through ELISA kits. The homeostasis model assessment of insulin resistance index (HOMA-IR) was calculated using the following equation: HOMA-IR = FBG × insulin/22.5 [16].

### 2.4. Determination of Biochemical Indices

The serum levels of TC, TG, LDL-C, and HDL-C were determined using the detection kits, which were used according to the manufacturer’s instructions.

### 2.5. Pathological Analysis

The pancreatic and skin tissues were fixed in a 4% paraformaldehyde solution for 48 h at room temperature, then embedded in paraffin and sectioned. The pancreatic sections were stained using H&E staining, and the skin tissues were stained using a combination of H&E and Masson’s trichrome.

### 2.6. Metabolomics Analysis

Before the metabolomics analysis, serum samples were subjected to metabolite extraction using the established methods [17]. The resulting supernatant was transferred to sample vials for LC–MS/MS analysis. The samples were analyzed in the Thermo UHPLC-Q Exactive HF-X system equipped with an ACQUITY HSS T3 column (Waters Corporation, Milford, CT, USA) at Majorbio Bio-Pharm Technology Co. Ltd. (Shanghai, China). Mobile phase A consisted of 0.1% formic acid in water–acetonitrile (95:5, *v*/*v*), and mobile phase B was acetonitrile–isopropanol (1:1, *v*/*v*) with 0.1% formic acid. Chromatography was carried out based on the standard protocol of Majorbio Bio-Pharm Technology Co., Ltd. (Shanghai, China). The pretreatment of LC/MS raw data was performed using Progenesis QI (version 2.3, Waters Corporation, Milford, CT, USA) software. Meanwhile, the metabolites were identified by searching the databases, and the central databases were the HMDB (http://www.hmdb.ca/), Metlin (https://metlin.scripps.edu/), and the Majorbio Database. Then, the data matrix obtained after searching the databases was uploaded to the Majorbio cloud platform (https://cloud.majorbio.com) for data analysis.

### 2.7. Statistical Analysis

Statistical analysis was performed using SPSS 21.0 software (SPSS Inc., Chicago, IL, USA) and Prism 8.3.0 (GraphPad Software, San Diego, CA, USA). Variables from multiple groups were compared using ANOVA, and several comparisons were performed using the Duncan algorithm (* *p* < 0.05, ** *p* < 0.01, and *** *p* < 0.001). Metabolomics data were analyzed on the Majorbio cloud platform (www.majorbio.com).

## 3. Results

### 3.1. Effects of MCS on Glucose Homeostasis, Lipid Levels, Chronic Systemic Inflammation, and Pancreatic Histology

This study evaluated the effect of MCS on the FBG of T2DM mice (Table 1). Compared to the NC group, the FBG of diabetic mice (DC group) was elevated significantly, but MCS and metformin treatments reversed these changes. After treatment for four weeks, the FBG levels of mice in the MMCS, HMCS, and MET groups were reduced by 49.65%, 34.79%, and 41.29%, respectively (Table 1). These results indicated that the high-dose MCS treatment (200 mg/kg) could reduce the FBG effectively. Insulin resistance and insulin secretion deficiency could result in hyperglycemia in T2DM. Compared to the NC group, the serum insulin levels of the DC group were very low, indicating that the T2DM mice exhibited abnormal insulin secretion (Figure 1A). MCS treatment significantly elevated serum insulin levels, particularly in the MMCS and HMCS groups (Figure 1A). The HOMA-IR method was extensively employed to assess insulin sensitivity. A decrease in the HOMA-IR value indicated an improvement in insulin sensitivity [16]. The HOMA-IR values were significantly higher in the DC group than in the NC group. Still, MCS and metformin treatments were able to reduce the HOMA-IR levels to varying degrees (Figure 1B).

OGTT was used to evaluate the blood glucose homeostasis in the mice groups [18]. The fluctuations in the blood glucose levels after oral glucose administration within 2 h are depicted in Figure 1C. The blood glucose levels peaked in each group within 30 min and then gradually decreased. The blood glucose level of the DC group was higher than other groups at each time point. After 2 h, the blood glucose level of the DC group was 20.68 mmol/L. The glucose levels of HMCS and MET groups decreased to 13.87 mmol/L and 16.25 mmol/L, respectively (Figure 1C). Furthermore, the OGTT area under the curve (AUC) was studied to estimate the glucose tolerance of mice (Figure 1D) [8]. The AUC of the DC group notably increased compared to the NC group, which indicated that the β cells in the pancreas and the blood glucose regulation ability of T2DM mice were damaged severely. The glucose tolerance of diabetic mice showed improvements across all interventions, with a notable enhancement observed in the HMCS and MET groups (Figure 1D). Thus, MCS supplementation can reduce the blood glucose level.

Multiple studies have indicated that hyperlipidemia and dyslipidemia are prevalent lipid disorders in individuals with diabetes [8,9]. Compared to the NC group, the TC, TG, and LDL-C levels of the DC group notably increased, while the HDL-C levels were decreased (Figure 2). After four weeks of treatment, the TC, TG, and LDL-C levels decreased, and the HDL-C level increased significantly in all the treated groups, indicating that MCS could alleviate dyslipidemia (Figure 2). In particular, the TC levels were reduced by 67.85% in the HMCS group, greater than the decrease observed in the MET group (57.14%) (Figure 2B). Furthermore, it has been suggested that obesity and T2DM may be associated with chronic low-grade inflammation [19]. Hence, we analyzed serum LPS levels in mice and found that they were significantly elevated in the DC group compared to the NC group. In contrast, serum LPS was reduced considerably in the MMCS and HMCS groups (Figure 3).

The islets of Langerhans, clusters of endocrine cells dispersed throughout the pancreas, are depicted in Figure 4 to show the effects of MCS on the histopathology of the pancreas. The pancreatic islets of the NC group exhibited normal cells with a well-defined structure, even distribution, tight organization, clearly visible, and no apparent pathological changes. In contrast, the islets in the DC group showed severe atrophy, blurred boundaries, irregular morphology, necrosis, and vacuolization (Figure 4). After treatment with MCS and metformin, pancreatic tissue damage was reduced to varying extents, especially with the MMCS, HMCS, and MET interventions, resulting in the islets nearly returning to their normal state. These findings were in agreement with a previous publication showing that MCS was beneficial in restoring impaired insulin-producing function [8].

### 3.2. MCS Promotes Wound Healing in T2DM Mice

To explore the impact of MCS on diabetic wounds, the healing process and wound areas were assessed. As shown in Figure 5A, the wounds gradually reduced over time in all groups. After three days of treatment, a notable difference was observed, with a more significant reduction in wound size in the NC (48.5%), HMCS (39.8%), and MMCS groups (38.3%), while others did not show significant effects on the wound size (Figure 5B). By day 10, no wounds were visible in the NC and HMCS groups. The percentage of wound closure on day 10 in the NC and HMCS groups reached 92.10% and 90.40%, respectively, surpassing the DC (84.20%) and MET groups (83.61%) (Figure 5C). Wound closure was notably enhanced in MCS-treated diabetic mice, particularly in the HMCS group. Interestingly, there was no significant difference between the MET and DC groups throughout the wound-healing process.

Histopathological changes in the skin wounds on day 10 were examined using the H&E and Masson trichrome staining, as depicted in Figure 5D, to further explore the impact of MCS on wound repair. In comparison to the NC group, the wounds in the DC group demonstrated diminished re-epithelialization, a trend similarly noted in the MET group. Conversely, neovascularization and complete wound re-epithelialization were evident in the NC and all MCS-treated groups (Figure 5D). After MCS treatments, collagen deposition was assessed using Masson trichrome staining (Figure 5D). Wounds of the NC and MCS-treated groups displayed densely packed collagen fibers with a parallel arrangement, particularly notable in the NC and HMCS groups. In contrast, the collagen fibers appeared irregular in the DC and MET groups (Figure 5D). MCS enhanced skin regeneration and reduced visible scarring in a dose-dependent manner relative to the DC and MET groups.

### 3.3. Effect of MCS on Serum Metabolomics in Mice

Untargeted metabolomics analysis was conducted on the serum samples of the NC, DC, and HMCS groups to investigate the underlying mechanism of MCS in promoting wound healing. Quality control (QC) sample clustering was evaluated using principal component analysis (PCA) in positive and negative ion modes. The PCA score plot revealed a close clustering of the QC samples, suggesting consistent analytical conditions and high repeatability of the detection process. Moreover, the analysis indicated a clear serum metabolite separation between the NC and DC groups, with HMCS showing closer proximity to NC (Figure 6A). The partial least-squares discriminant analysis (PLS-DA) demonstrated significantly different metabolite profiles among NC, DC, and HMCS in negative and positive ion modes (Figure 6B). The metabolite profiles showed good separations in the serum samples of the NC, DC, and HMCS groups, indicating that STZ and MCS treatments could cause changes in the serum biomarkers. Moreover, the model’s accuracy was assessed through permutation testing, and the results illustrated in Figure 6C show the R^2^ (0.0994, 0.099) and Q^2^ (−0.551, −0.641) values, which reveal that the PLS-DA models exhibited good repeatability and stability for the metabolomics study.

To obtain the biomarkers in T2DM mice, we used orthogonal partial least-squares discrimination analysis (OPLS-DA) to examine the differences in serum metabolites among the NC, DC, and HMCS groups. After the analysis, we obtained a total of 570 metabolites with changes in the DC group, of which 361 were upregulated and 209 were downregulated (Figure 7A). HMCS exhibited a distinct metabolite profile compared to the DC group, with 232 metabolites showing significant differences, including 57 upregulated and 175 downregulated metabolites (Figure 7B). To further examine the metabolic pathways involved in the metabolites, the metabolite data that had been significantly altered were compared with the Kyoto Encyclopedia of Genes and Genomes (KEGG) database to identify the enriched pathways. As illustrated in Figure 7C,D, the intensity of the circle’s color corresponds to the −log *p* value, indicating the significance of the metabolic pathway. Additionally, the size of the circle denotes the relative impact of the pathway [20]. The KEGG database revealed that there were 10 main pathways identified between the DC and NC groups and 10 distinct pathways between the DC and HMCS groups. The differential metabolites among the NC, DC, and HMCS groups were analyzed, and five KEGG pathways were selected. These pathways include glycerophospholipid metabolism, fructose and mannose metabolism, steroid hormone biosynthesis, pyrimidine metabolism, and the Krebs cycle. The metabolites associated with these pathways include glycerol-3-phosphate (Gro3P), cortisol, L-rhamnose, urea, 2′-deoxyuridine, and malic acid.

## 4. Discussion

T2DM is a chronic disease characterized by hyperglycemia with complications, such as diabetic foot ulcers, kidney diseases, and eye diseases [21]. Saponins, the primary constituents of bitter melon, have demonstrated efficacy in enhancing glucose homeostasis and reducing insulin resistance [8]. In this study, MCS effectively improved glucose metabolism and insulin secretion, consistent with previous studies. The HOMR-IR index can be utilized to assess the degree of insulin resistance in T2DM mice. A higher HOMA-IR value indicates greater insulin resistance [16]. Another prominent feature of diabetes is lipid metabolic disorders. In general, an increase in TC, TG, and LDL-C levels and a decrease in HDL-C levels were observed. This study found that MCS could decrease TC, TG, and LDL-C and increase HDL-C levels. Overall, MCS reduced blood glucose and lipids in T2DM mice.

Wound healing requires infection management, inflammation reduction, connective tissue matrix regeneration, vasculogenesis promotion, wound closure, and re-epithelialization. Diabetic patients exhibit irregular wound healing patterns, as their wounds encounter challenges in healing due to peripheral vascular disease, sensory-motor impairment, autonomic neuropathy, and other factors [10,11]. Bitter melon extracts have been shown to improve wound healing by stimulating angiogenesis, collagen fiber synthesis, and fibroblast proliferation, as well as promoting granulation tissue formation, indicating their potential as an external therapeutic agent [10,11]. However, the potential of MCS to enhance diabetic wound healing through oral administration is also noteworthy. In this study, the results showed that MCS increased wound epithelial regeneration, collagen fiber deposition, neovascularization, and granulation tissue formation through oral administration.

Metabolomics, a systems biology technique, offers comprehensive metabolic insights about biological samples, enhancing our understanding of pathological mechanisms and facilitating advancements in prediction, early diagnosis, and treatment [13]. Diabetes is a chronic condition characterized by persistent metabolic irregularities, which has garnered significant interest among researchers employing metabolomics for in-depth studies. The present study identified 570 metabolites expressed significantly differently between the diabetic and normal controls. Further analysis revealed that six potential biomarkers in the NC and DC groups were significantly affected after treatment with HMCS. The enriched metabolites, such as Gro3P, cortisol, L-rhamnose, and malic acid in the DC group, were significantly reduced after MCS treatment, while the expression of urea and 2′-deoxyuridine was upregulated.

Gro3P is an integral component of glucose, lipid, and energy metabolism in mammalian cells and is believed to participate in glycolysis, gluconeogenesis, and lipid synthesis. Gro3P serves as a primary substrate for lipid synthesis [22]. Through the action of glycerol 3-phosphate acyltransferase, Gro3p forms aggregates with long-chain acyl-coenzyme A to generate lysophosphatidic acid [23]. In this study, Gro3P was downregulated after the MCS treatment. The downregulation of Gro3P can reduce the efficiency of glycerophospholipid and glycerolipid pathways and TG production, thereby inhibiting lipid accumulation. The metabolism of glycerophospholipids is crucial for the proper functioning of substance and energy metabolism, as well as for maintaining metabolic balance and homeostasis. This study showed that MCS had an evident callback effect on the glycerol phospholipid metabolism disorder in T2DM mice.

The characteristics of glucose metabolism disorder in high-risk patients with diabetes are apparent. The experimental results showed that compared to NC, the levels of glucose and other sugars (L-rhamnulose) in the serum of the DC group showed a significant upward trend. D-glucose and L-rhamnulose are involved in fructose and mannose metabolism, which belongs to carbohydrate metabolism. Fructose and mannose are isomers of glucose, which can be converted to glucose through enzymatic action, thereby entering the glycolytic process. Chronic tissue inflammation is thought to be caused by abnormal fructose and mannose metabolism, indicated by increased peripheral blood monocytes, decreased bone marrow monocyte activity, and the infiltration of tissue inflammatory factors [24]. Cortisol is an adrenal glucocorticoid hormone that plays a role in the metabolism of proteins, carbohydrates, and other substances. It also helps maintain normal physiological functions in the body and indicates the level of the stress response [25]. Excessive release of cortisol can lead to platelet adhesion, thrombosis, endometrial thickening, and disruption of the body’s oxidative balance, hindering wound healing. In addition, excessive secretion of cortisol can enhance the function of the pituitary–adrenal cortex system and exacerbate stress responses. The adrenal cortex can activate inflammatory factors such as lymphocytes, macrophages, and granulocytes, aggravating the inflammatory response [26]. MCS can diminish serum concentrations of glucose, rhamnose, and cortisol in T2DM mice, alleviating body inflammation and consequently enhancing wound healing in these mice.

Deoxyuridine, a differential metabolite, plays a crucial role in pyrimidine metabolism and can be converted into deoxyribose 1-phosphate and uracil by thymidine phosphorylase [27]. Uracil can be decomposed into N-carbamoyl-β-Alanine and then passed through β-Ureapropionase, further decomposing into β-Alanine, CO_2_, and ammonia. β-alanine and β-aminoisobutyric acid are -NH_2_ donors for the transamination of α-ketoglutaric acid to glutamate, ultimately metabolizing to produce urea, H_2_O, and CO_2_ [28]. Serum urea, a byproduct of protein and amino acid metabolism in the body, serves as an indicator for assessing the equilibrium between protein breakdown and synthesis [29]. A lower blood urea level indicates a decreased effect of protein and amino acid catabolism, leading to a decline in physical performance in the body. The results showed that the contents of deoxyuridine and urea in T2DM mice were significantly reduced compared to the NC group and that the MCS treatment could partly recover the level of these metabolites.

The Krebs cycle is the major energy-producing pathway, through which fatty acids, glucose, and amino acids are synthesized and transformed. The regular operation of the Krebs cycle can comprehensively reflect the body’s energy metabolism level. Malic acid is the primary intermediate metabolite in the Krebs cycle, and the level of malic acid impacts the Krebs cycle [30]. This study showed a notable increase in malic acid levels within the DC group, which decreased following MCS intervention. From these analyses, MCS could regulate the abnormality of glucose, lipid, amino acid, and energy metabolism and reduce body inflammation, thereby improving glucose homeostasis and promoting wound healing.

## 5. Conclusions

Glucose and lipid metabolism disorder, inflammation, oxidative stress, and apoptosis are essential factors in the onset and progression of diabetic foot ulcers. MCS could reduce glucose and lipid metabolism disorders, improve insulin resistance, and promote the epithelial regeneration rate and collagen fiber deposition. Moreover, we also found that MCS could affect many metabolism pathways, including lipid, carbohydrate, amino acid, and energy metabolism, to exert therapeutic efficacy. The findings suggest that MCS exhibits encouraging wound-healing properties in a diabetic animal model via multiple pathways. Nevertheless, it is crucial to emphasize that clinical trials are essential to confirm MCS’s efficacy and safety in human subjects.

## Figures and Tables

**Figure 1 foods-13-03163-f001:**
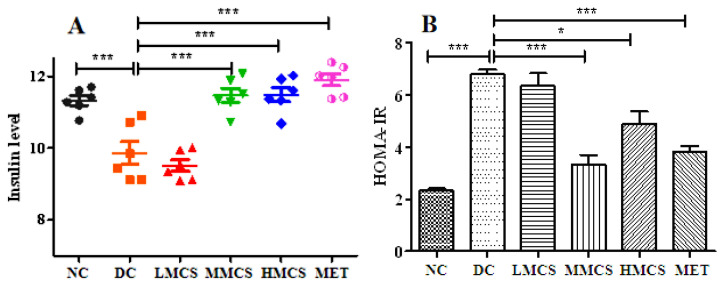
Effect of MCS on insulin levels (**A**), HOMA-IR (**B**), OGTT (**C**), and AUC (**D**) in type 2 diabetic mice (n = 6). * *p* < 0.05, ** *p* < 0.01, *** *p* < 0.001, versus the diabetic control group.

**Figure 2 foods-13-03163-f002:**
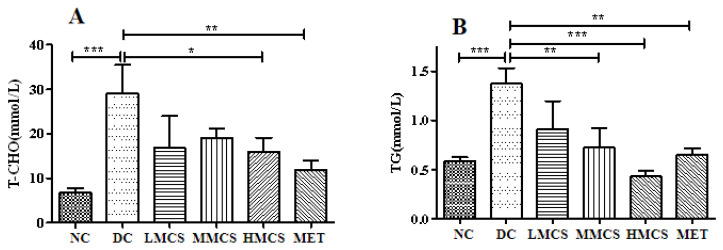
Effects of MCS on TC (**A**), TG (**B**), LDL-C (**C**), and HDL-C (**D**) levels in T2DM mice (n = 6). * *p* < 0.05, ** *p* < 0.01, *** *p* < 0.001, versus the diabetic control group.

**Figure 3 foods-13-03163-f003:**
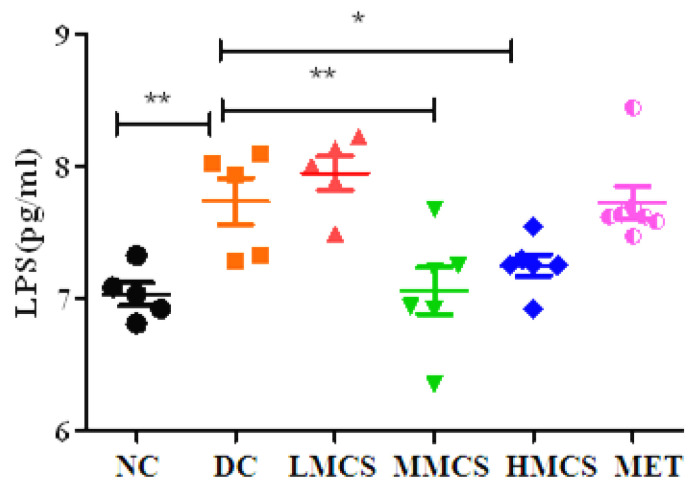
Effect of MCS on the serum LPS in T2DM mice (n = 6). * *p* < 0.05, ** *p* < 0.01, versus the diabetic control group.

**Figure 4 foods-13-03163-f004:**
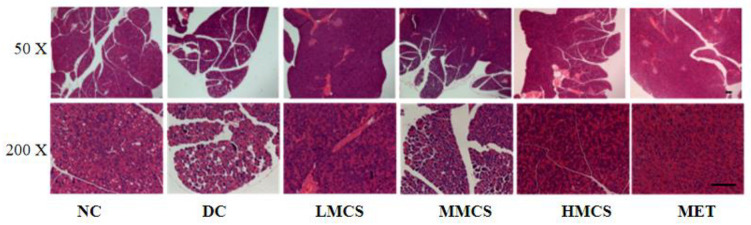
Effects of MCS on the morphology of pancreas in T2DM mice. (scale bar: 200 µm).

**Figure 5 foods-13-03163-f005:**
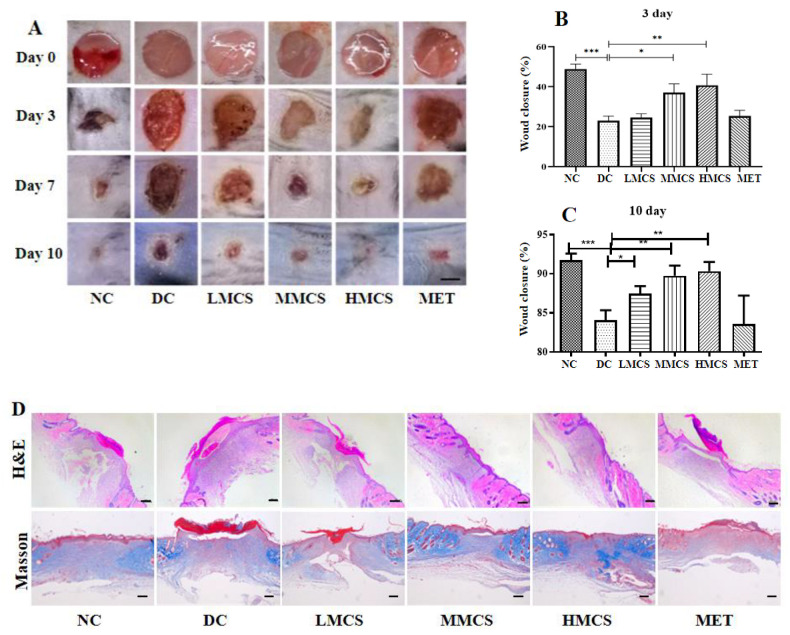
Wound healing progress: images of a representative wound site on post-injury days (**A**) (scale bar: 2 mm), the effects of MCS treatments on the wound closure rate on day 3 (**B**) and day 10 (**C**), H&E staining and Masson’s trichrome staining of wounds at day 10 (**D**) (scale bar: 200 µm) (n = 6). * *p* < 0.05, ** *p* < 0.01, *** *p* < 0.001, versus the diabetic control group.

**Figure 6 foods-13-03163-f006:**
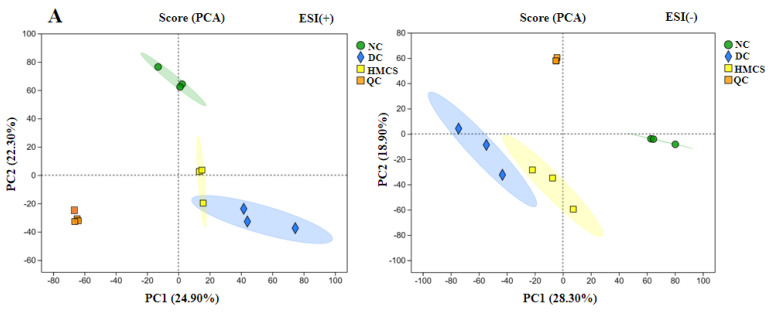
PCA score plots (**A**), PLS-DA score plots (**B**), and PLS-DA validation plot intercepts (**C**) (n = 3).

**Figure 7 foods-13-03163-f007:**
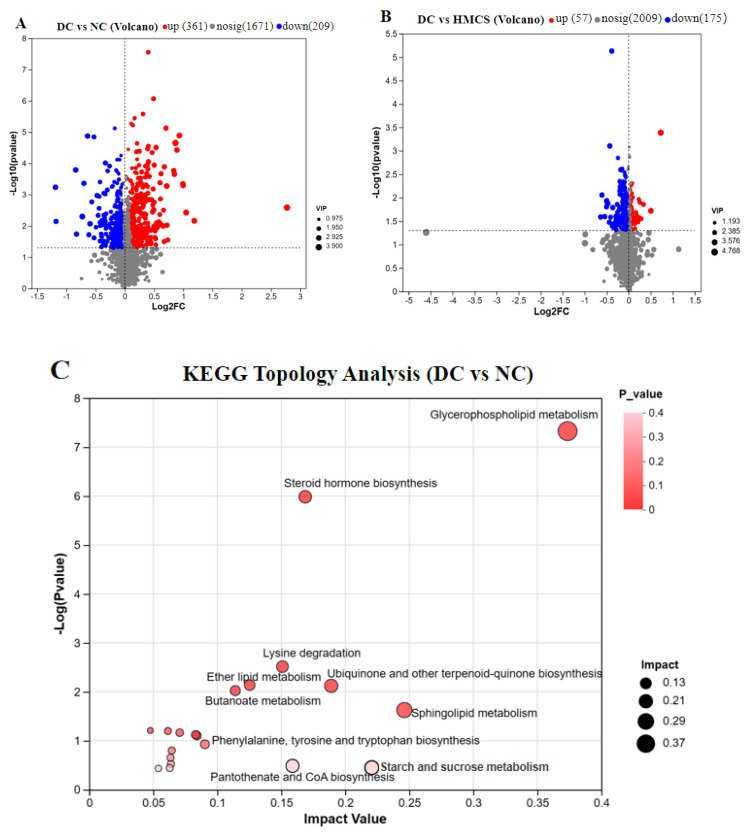
Effects of MCS on serum metabolic profiles of T2DM mice. Volcano plot analysis: DC versus NC (**A**) and DC versus HMCS (**B**), KEGG metabolic pathways enrichment analysis: DC versus NC (**C**) and DC versus HMCS (**D**), the main differential metabolites (**E**–**J**) (n = 3). * *p* < 0.05, ** *p* < 0.01, *** *p* < 0.001.

**Table 1 foods-13-03163-t001:** Dynamic FBGs (mmol/L) in different groups.

Group	Day 1	Day 7	Day 14	Day 21	Day 28
NC	4.71 ± 0.83 ^a^	4.27 ± 0.41 ^a^	4.16 ± 0.85 ^a^	4.78 ± 0.32 ^a^	4.88 ± 0.65 ^a^
DC	14.54 ± 1.99 ^b^	19.28 ± 5.40 ^b^	15.54 ± 0.84 ^b^	17.64 ± 2.15 ^b^	15.58 ± 1.02 ^b^
LMCS	14.65 ± 1.58 ^b^	17.90 ± 2.00 ^b^	12.84 ± 2.86 ^b^	14.14 ± 2.80 ^c^	13.66 ± 1.36 ^c^
MMCS	14.30 ± 1.12 ^b^	14.82 ± 1.21 ^b^	12.64 ± 1.76 ^b^	11.36 ± 0.96 ^c^	7.20 ± 0.48 ^d^
HMCS	14.60 ± 1.92 ^b^	17.67 ± 1.93 ^b^	13.56 ± 2.59 ^b^	12.48 ± 1.72 ^c^	9.52 ± 1.54 ^e^
MET	14.58 ± 2.84 ^b^	19.16 ± 1.78 ^b^	12.70 ± 2.48 ^b^	12.40 ± 2.67 ^c^	8.56 ± 0.23 ^de^

Note: Data are presented as the mean ± SD, different letter superscripts indicate significant differences among groups on the same day (*p* < 0.05). Abbreviations: NC, normal control; DC, diabetic control; LMCS: low-dose MCS; MMCS: medium-dose MCS; HMCS: high-dose MCS; MET: metformin.

## Data Availability

The original contributions presented in the study are included in the article, further inquiries can be directed to the corresponding author.

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
