# Peer review of "Metabolomics Revealed the Effects of Momordica charantia L. Saponins on Diabetic Hyperglycemia and Wound Healing in Mice"

_foods, 2024, doi:10.3390/foods13193163_

Round 1

Reviewer 1 Report

Comments and Suggestions for Authors

In this study, the postprandial metabolomic (LC-MS), antihyperglycemic (OGTT), anti-inflammatory (LPS), antilipidemic (cholesterol/TAG/lipoproteins) and wound healing effects (as related to diabetic foot) of a saponin-rich bitter melon (Mormodica charantia) extract (MCS) were systematically and sequentially evaluated in diabetic mice (STZ-induced Male C57BL/6J mice). Histological changes in pancreas and skin were also evaluated. The presented evidence accounts for the positive effects on these physiological parameters. Although the experimental design/execution were impeccable, there is still room for improvement in the discussion, conclusions and format of the manuscript. Here are some observations that may help to improve the study´s scientific soundness and uniqueness.

General. A) The reading and comprehension of the manuscript will improve furthermore if it is independently reviewed by a native English-spoken colleague or by a formal translation agency. B) Besides scientific names, use italics when appropriate. C) The meaning of each acronym or abbreviation used must be defined the first time it is cited and, if possible, reduce them as possible (e.g. DFUs is not needed)

Title/Abstract/Introduction/Methods. OK.

Results & Discussion. A) Eliminate discussion-like statements within results section. B) Try as much as possible making comparisons with preceding studies on this topic. C) Avoid making assertions without evidence-based support.

Tables and figures: A) The images must be supplied at a better resolution (>300 dpi) while tables should be formatted according to Foods´ guidelines. B) Do not use abbreviations in titles and footnotes, otherwise include their meaning.

References. A) It is recommended to review again as some do not have the appropriate format.

Comments on the Quality of English Language

Minor review. English grammar and syntax should be reviewed a formal translation agency or by a native English-speaking colleague.

Author Response

Reviewer #1: In this study, the postprandial metabolomic (LC-MS), antihyperglycemic (OGTT), anti-inflammatory (LPS), antilipidemic (cholesterol/TAG/lipoproteins) and wound healing effects (as related to diabetic foot) of a saponin-rich bitter melon (Mormodica charantia) extract (MCS) were systematically and sequentially evaluated in diabetic mice (STZ-induced Male C57BL/6J mice). Histological changes in pancreas and skin were also evaluated. The presented evidence accounts for the positive effects on these physiological parameters. Although the experimental design/execution were impeccable, there is still room for improvement in the discussion, conclusions and format of the manuscript. Here are some observations that may help to improve the study´s scientific soundness and uniqueness.

 1.General. A) The reading and comprehension of the manuscript will improve furthermore if it is independently reviewed by a native English-spoken colleague or by a formal translation agency. B) Besides scientific names, use italics when appropriate. C) The meaning of each acronym or abbreviation used must be defined the first time it is cited and, if possible, reduce them as possible (e.g. DFUs is not needed)

Response: The manuscript has been reviewed by a formal translation agency again and the modified parts have been highlighted in red. Other comments has been revised as suggestion of the reviewer.

  1. Title/Abstract/Introduction/Methods. OK.

Response: Thanks for your review.

  1. Results & Discussion. A) Eliminate discussion-like statements within results section. B) Try as much as possible making comparisons with preceding studies on this topic. C) Avoid making assertions without evidence-based support.

Response: It has been revised as suggestion of the reviewer. Some of results & discussion have been rewritten.

  1. Tables and figures: A) The images must be supplied at a better resolution (>300 dpi) while tables should be formatted according to Foods´ guidelines. B) Do not use abbreviations in titles and footnotes, otherwise include their meaning.

 Response: It has been revised as suggestion of the reviewer.

  1. A) It is recommended to review again as some do not have the appropriate format.

Response: References has been review again and some of them has been changed.

  1. Comments on the Quality of English LanguageMinor review. English grammar and syntax should be reviewed a formal translation agency or by a native English-speaking colleague.

Response: The manuscript has been reviewed by a formal translation agency again.

Reviewer 2 Report

Comments and Suggestions for Authors

Dear sirs/madams

I've been glad to review your paper. It is, in my opinion, a real good work about an  interesting topic, well structured and presented.
I do not have suggestions to improve or modify your paper, except for minor checks regarding  different font used in some parts and check the correct allignement of captions in figures.

My compliments for your excellent work.

Author Response

Reviewer #2: I've been glad to review your paper. It is, in my opinion, a real good work about an  interesting topic, well structured and presented.
I do not have suggestions to improve or modify your paper, except for minor checks regarding  different font used in some parts and check the correct allignement of captions in figures. My compliments for your excellent work.

Response:Thanks for your review.

Reviewer 3 Report

Comments and Suggestions for Authors

Title: Metabolomics revealed the therapeutic effects of Momordica charantia saponins on diabetic hyperglycemia and wound healing in mice.

The title of the manuscript is consistent with the topic of the study. The research presented in this article examines the effects and mechanisms of MCS on diabetic wounds. Results have shown that higher Momordica charantia L. saponins (MCS) intake lowered fasting blood 15 glucose levels, serum lipids, and lipopolysaccharides in diabetic mice. According to the authors; diabetic mice treated with MCS exhibited faster wound healing than the diabetic control groups. After three days, the diabetic control groups demonstrated a wound area reduction of only 19.3%, whereas a 39.75% reduction of wound reduction was seen after high-dose treatment of MCS. Additionally, in the metabolomics study, six potential biomarkers were screened. The results revealed that MCS mainly regulated glycerophospholipid metabolism, fructose and mannose metabolism, steroid hormone biosynthesis, pyrimidine metabolism, and the TCA cycle (Krebs cycle) to affect wound healing.

The work is quite detailed and precise but contains a few minor errors, both linguistic and factual. In the section of the discussion, the authors draw constructive conclusions. The scope of literature data is up-to-date and consistent with the subject of the research undertaken.

Comments and suggestions for Authors:

·         In the Abstract the Authors use the abbreviation TCA cycle, please clarify whether this is about the Krebs cycle (Citric acid cycle) or Tricarboxylic acid cycle. Because in the abbreviations at the end of the manuscript, there is TCA as tricarboxylic acid. In my opinion, it is misleading.

·         The Introduction section should be expanded with information on why we should lean towards using natural raw materials rather than synthetic drugs for diabetic wounds.

·         In the section Materials and Methods in chapter 2.1. (line 78), the Authors give information about chemical reagents. In my opinion, the Authors should specify what chemical reagents they used for the research.

·         In my opinion, the Authors should remove the "Figure Captions" section and caption each figure directly.

In my opinion, the manuscript needs minor corrections.

Author Response

Reviewer #3: The title of the manuscript is consistent with the topic of the study. The research presented in this article examines the effects and mechanisms of MCS on diabetic wounds. Results have shown that higher Momordica charantia L. saponins (MCS) intake lowered fasting blood 15 glucose levels, serum lipids, and lipopolysaccharides in diabetic mice. According to the authors; diabetic mice treated with MCS exhibited faster wound healing than the diabetic control groups. After three days, the diabetic control groups demonstrated a wound area reduction of only 19.3%, whereas a 39.75% reduction of wound reduction was seen after high-dose treatment of MCS. Additionally, in the metabolomics study, six potential biomarkers were screened. The results revealed that MCS mainly regulated glycerophospholipid metabolism, fructose and mannose metabolism, steroid hormone biosynthesis, pyrimidine metabolism, and the TCA cycle (Krebs cycle) to affect wound healing.

The work is quite detailed and precise but contains a few minor errors, both linguistic and factual. In the section of the discussion, the authors draw constructive conclusions. The scope of literature data is up-to-date and consistent with the subject of the research undertaken.

Comments and suggestions for Authors:

1.In the Abstract the Authors use the abbreviation TCA cycle, please clarify whether this is about the Krebs cycle (Citric acid cycle) or Tricarboxylic acid cycle. Because in the abbreviations at the end of the manuscript, there is TCA as tricarboxylic acid. In my opinion, it is misleading.

Response: It has been revised as suggestion of the reviewer.

2.The Introduction section should be expanded with information on why we should lean towards using natural raw materials rather than synthetic drugs for diabetic wounds.

Response: It has been revised as suggestion of the reviewer, showing in line 37-46.

3.In the section Materials and Methods in chapter 2.1. (line 78), the Authors give information about chemical reagents. In my opinion, the Authors should specify what chemical reagents they used for the research.

Response: The use of chemical reagents was added as suggestion of the reviewer

4.In my opinion, the Authors should remove the "Figure Captions" section and caption each figure directly

Response: It has been revised as suggestion of the reviewer.

5.I think that in the title should be "Momordica charantia L." and the word "saponins" without italic.

Response: It has been revised as suggestion of the reviewer.

6.Please reduce the spacing in the text here.

Response: It has been revised as suggestion of the reviewer.

7.Specify that this is about the Krebs cycle (TCA cycle), please

Response: It has been revised as suggestion of the reviewer, and rewriting the manuscript.(Line347-351)